

# Plants in movement – Floristic and climatic characterization of the New Jersey hinterland during the Palaeogene-Neogene transition in relation to major glaciation events

Sabine Prader[1,2],*, Ulrich Kotthoff[1,2], Francine M.G. McCarthy[3], Gerhard Schmiedl[4,2],
Timme H. Donders[5] and David R. Greenwood[6]

1Center for Natural History, Hamburg University, Bundesstraße 55, D-20146 Hamburg, Germany
2Institute of Geology, University of Hamburg, Bundesstrasse 55, D-20146 Hamburg, Germany
3Department of Earth Sciences, Brock University, 1812 Sir Isaac Brock Way, St. Catharines, Ontario, L2S 3A1, Canada
4Center for Earth System Research and Sustainability, Hamburg University, Bundesstraße 55, D-20146 Hamburg, Germany
5Palaeoecology, Department of Physical Geography, Heidelberglaan 2 3584 CS Utrecht, The Netherlands
6Department of Biology, Brandon University, 270 18th Street, Brandon, Manitoba, R7A 6A9, Canada

*Correspondence to*: Sabine Prader (sabine.prader@uni-hamburg.de)

**Abstract.** Mid-Oligocene to Early Miocene terrestrial palynomorphs from the New Jersey hinterland (eastern North America: IODP-Expedition 313) were analysed, using light microscopy and scanning electron microscopy, to infer altitudinal spatial and long-term temporal vegetation migration in context of global climate change. The mesophytic forest was the most widespread vegetation type in the hinterland, with *Quercus* (Group Quercus, Quercus/ Lobatae and aff. Group Protobalanus) being the dominant taxon. Pollen grains of the extinct genus *Eotrigonobalanus* (Fagaceae) are documented.

To infer possible topographic palaeovegetation movements during the selected time interval terrestrial palynomorphs were assigned to six vegetation units. Relative abundances of vegetation units show weak temporal and spatial fluctuations, with the sum of bisaccate pollen grains being most pronounced. Periodic changes in vegetation units suggest movements of the plant cover responding to orbital-scale glacial-interglacial changes of the Oligocene and early Miocene. Relative abundances of several taxa (e.g. *Carya*) did not change significantly during the Oligocene, but alterations are recognizable when compared with an already published late Middle Miocene record from the same area, probably indicating biotic responds to environment change. A pollen-based bioclimatic analysis with four standard parameters (mean annual temperature, mean temperatures of the coldest and warmest month, mean annual precipitation) was performed to reconstruct palaeoclimatic changes indicating weak fluctuations in temperature and precipitation.

## 1.Introduction

At the Eocene-Oligocene boundary (EOB) at ~33.7 Ma global climate began to transform from a greenhouse to an icehouse state (Zachos, et al., 2001; Pagani et al., 2005; Eldrett et al., 2009). Various mechanisms are under debate as to what caused the large-scale glaciation of Antarctica around the first Oligocene isotopic event (Oi-1 event), including the opening of the Drake Passage (Scher and Martin, 2006), the decline of atmospheric carbon dioxide concentrations (Pagani et al., 2005), and



orbital forcing (Coxall et al, 2005). The Oligocene is marked by a series of eight unipolar glaciation periods (Pekar et al., 2002), during which Antarctica experienced ice sheet growth, resulting in large glacioeustatic sea level fluctuations (Pekar et al., 2002; Wade and Pälike, 2004; Pälike et al., 2006; Pekar et al, 2006). Waxings and wanings of Eastern Antarctic ice sheets (EAIS; Zachos et al., 2001) during the Oligocene were not all of the same magnitude. The strongest glacial episode during this

epoch is represented by the Oi-2b event (27.0 Ma – 26.6 Ma; Pekar et al., 2006) (Pälike et al., 2006), which resulted in a glacioeustatic lowering of ~45 m (Pekar et al., 2002). These climate oscillations are attributed to insolation changes associated with orbital forcing (Wade and Pälike 2004; Pälike et al., 2006).

The inception of the Miocene isotopic event 1 (Mi-1 event), the second largest climatic aberration since the Oi-1 event (Lear et al., 2004), coincides with the Oligocene-Miocene boundary (OMB) at 20.03 Ma (Liebrand et al., 2011). It represents the

first and largest cooling episode during the Miocene with a sea-surface temperature decline of ~ 2 C° occurring prior the event (Lear et al., 2004). Mechanisms behind this global cooling event are probably related to late Oligocene atmospheric carbon dioxide levels being near to those of modern times (Pagani et al., 2005; Roth-Nebelsick et al., 2014) and orbital forcing (Zachos et al., 2001; Pälike et al., 2006).

From a terrestrial perspective, the equator to pole climatic gradient became more pronounced since the Oi 1 global event,

resulting in a predominantly temperate climate in the mid latitudes of the Northern Hemisphere (Hably et al., 2000). This time interval was important for a renewal of the North American (Graham, 1999) and other floras (e.g. Eldrett et al., 2009). Most records of early Oligocene age from the mid latitudes of Europe indicate a prevailing deciduous vegetation character with intermixed Eocene floristic elements like *Eotrigonobalanus* (e.g. Velitzelos et al., 2014). Only a few early Oligocene records imply highly diverse broad-leaved macrofloristic assemblages in the mid latitudes (Kovar-Eder, 2016). Terrestrial records of

the Southern Hemisphere indicate an increase of temperate tree taxa in the fossil record too, with *Nothofagus* being the dominant taxon (e.g. Prebble et al., 2017).

Most information about the floristic and climatic history of the Oligocene of North America is derived from western (e.g. Meyer and Manchester, 1997; Grímsson et al., 2016) and interior deposits (e.g. Wolfe and Schorn, 1989; 1990) of the continent. For the early Oligocene, records from the western part of North America document a dominance of temperate climates,

promoting the growth of deciduous forests with a small portion of broadleaf evergreens in the understorey (Meyer and Manchester, 1997) and persistent single warmth-loving elements of the Eocene (Grímsson et al., 2016). At the same time, intensifying seasonality triggered the development of open landscapes and dry conifer woodlands in the interior of the country (Wolfe and Schorn, 1989; 1990). By the end of the Oligocene, the prevailing climate determined the formation of open grasslands in the western interior of North America (Retallack, 2004; Strömberg, 2005).

The palaeofloristic and terrestrial environment of eastern North America is not well understood due to a scarcity of fossil Oligocene floras (e.g. Frederiksen, 1991). Generally, terrestrial palynomorphs deposited in marine and continental sediments are well suited for the reconstruction of palaeoclimate (e.g. Eldrett et al., 2009), palaeovegetation (e.g. Kmenta and Zetter 2013) and biogeographic history (e.g. Grímsson et al., 2016). Sediment cores drilled in the framework of the Integrated Ocean Drilling Program (IODP) Expedition (Exp.) 313 to the New Jersey Shallow Shelf (NJSS) allowed for the reconstruction of





Oligocene and Miocene terrestrial ecosystem development in eastern North America. The recovered deposits contain generally well preserved marine and terrestrial palynomorphs. Furthermore, a robust age model (Browning et al., 2013) was developed for these cores. Kotthoff et al. (2014) gave an overview of the OMB, including the related Mi-1 cooling event indicated by temperature decline and conifer expansion. However, only a few records world-wide (e.g. Prebble et al., 2017) deal with the

response of plant cover to Oligocene global climatic oscillations. Because global climate changes do not affect all regions at the same scale, biotic responses may differ from one environment to another (Utescher et al, 2015). Climate models indicate more constant conditions for the Atlantic east coast during the Oligocene than for other regions (von der Heydt and Dijkstra, 2006), implying a restricted vegetation turnover for the hinterland.

In these contexts, we have evaluated the regional floristic behaviour of the NJSS hinterland to provide a more detailed insight

into a mid-latitude terrestrial system during the middle Oligocene and the Early Miocene and thus to contribute to a better understanding of patterns of plant persistence/migration.

## 2. Material and methods

### 2.1 Geological setting

#### 2.1.1 Site M0027: selection, age model and core sediment description

Our study focuses on sediments from Site M0027 recovered during IODP Expedition 313 to the New Jersey Shallow Shelf (NJJS). The drilling position of Hole M0027A is at 39°38.046'N and 73°37.301'W at 33.5 m water depth and at a site-shoreline distance of 40 km (Fig. 1). Site M0027 is the only of three recovered sites containing Oligocene sequences (Browning et al., 2013; Miller et al., 2013a, 2013b).

The age model is based on calcareous nannofossils, dinoflagellate cysts, planktic diatoms, Sr isotopes, and sequence

stratigraphy (Browning et al., 2013; Miller et al., 2013a). The analysed core sediments comprise four sequences, which correlate with sea level high stands, while no sediments were deposited during sea-level lowstands (likely connected to glacial phases). Sequence O3 (617 to 538.68 mbsf), the oldest analysed sequence, was deposited between ~29.3 to ~28.2 Ma (late Rupelian/Chattian; Fig. 2). This sequence contains two intra-sequences, which are tied to facies changes: Sequence o.1 at 596.3 mbsf (~29.0 Ma) and Sequence o.5 at 563.0 mbsf (~28.6 Ma; Miller et al., 2013a). Sequence O6 (538.68 to 509 or 515 mbsf,

see Supplement S1) comprises the uppermost part of the Oligocene and the transition to the early Miocene (Chattian/Aquitanian). The estimated age of sequence O6 is best dated to ~23.5 to ~23.0 Ma. The early Miocene Sequence m6 (509 or 515 to 494.87 mbsf) covers an age interval of ~20.9 to ~20.7 Ma (Aquitanian). Sequence m5.8 (494.87 to 361.28) is dated to ~20.1 to ~19.2 Ma (late Aquitanian/early Burdigalian). For more details on age model compare Supplement S1. All analysed sequences lie within Lithological Unit VII, comprising coarse-grained to fine-grained sands interbedded with silty

clay laminae (Expedition 313 Scientists, 2010).



Reconstructions by Scotese et al., (1988) imply that the NJSS study area was situated ~2° further south during the Oligocene and Miocene, and reached its modern position between 39° and 40°N during the Pliocene. The onset of significant topography change and significant increase in relief variation of the Appalachian Mountains started at ~20 Ma; prior to this time the Appalachians showed low modifications in relief and elevation (Liu, 2014).

## 2.2 Palynology

### 2.2.1 Sample processing and analyses

Sample processing was performed at Brock University, St. Catharines, Canada (sieving, HCl and HF-treatment) and at the Laboratory of University of Hamburg (acetolysis). Around 300-400 terrestrial palynomorphs were identified using mostly the acetolyzed treated material. For percentage calculations, bisaccate pollen grains were excluded from the reference sum as these

grains tend to be overrepresented in marine records (Mudie and McCarthy, 1994; Eldrett et al., 2009). Additionally, 300 marine/terrestrial palynomorphs were counted using the non-acetolyzed material due to sensitivity of some dinoflagellate cysts to acetolysis. The P:D ratio (pollen versus dinoflagellate cyst) is an indicator for transport mechanisms and sea-level fluctuations (McCarthy et al., 2003), where high P:D indicates low sea level and/or enhanced run off. In sum, 56 samples were analysed spanning the time interval of the middle late Oligocene to the early Miocene. Additionally, some pollen grains were

analyzed via SEM. We applied the Shannon-Wiener-Index ($H(s)$) in order to assess changes in diversity. More details on sample processing and analyses are given in Supplement S2.

### 2.3 Reconstruction of palaeovegetation and palaeoclimate

Terrestrial palynomorphs were grouped into six artificial palaeovegetation units: 1: high-altitude conifer forest; 2: mid-altitude conifer forest; 3: Cupressaceae; 4: mesophytic forest growing on well-drained soils; 5: mesophytic forest growing on moist/wet

soils; 6: Mesophytic understorey; 7: plant community associated to coastal environments, growing on sun-exposed sandbanks. Grouping was done according to growing preferences of modern analogues, including soil condition and altitudinal zonation (Supplement S3). This superordinate artificial grouping is congruent with the vegetation units of the late Mid-Miocene of the same area (Prader et al., 2017). The generalisation allows inference of shifts within the terrestrial vegetation.

Reconstructions of palaeoclimatic conditions used the bioclimatic analysis after Greenwood et al. (2005) and Prebble et al.

(2017), which is based on the nearest living relative concept (NLR). Differences to the likewise NLR-based Coexistence Approach (Utescher et al., 2014) are described in detail in Prebble et al. (2017).

Four different climatic parameters were generated as a standard to characterize the palaeomacroclimatic conditions: mean annual temperature (MAT), coldest month mean temperature (CMMT), warmest month mean temperature (WMMT), and mean annual precipitation (MAP). The generated palaeoclimatic estimates were based on climatic profiles of North American

and Chinese species (for details see Supplement S3). Table S3 summarizes the NLRs together with the climate source used.



## 3. Results

### 3.1 Terrestrial palynomorphs

Fair preservation of pollen grains allowed for identification of 72 taxa. Supplement S3 gives a summary of all identified terrestrial palynomorphs. Light Micrograph (LM) images of several well-preserved plant microfossils are shown on Plates S4-

i and S4-ii (Supplement S4), and Plate I illustrates SEM images of selected pollen grains. Within the 56 analysed samples, *Quercus* is the most abundant taxon, represented by three infrageneric groups: Group Quercus (Plate I C-D), Group Quercus/ Lobatae (Plate I: E-H) and aff. Group Protobalanus (Plate I: I-J). The number of *Quercus* pollen grains reaches > 50 % in most samples, except those at 572.15 mbsf and 570.02 mbsf (Fig. 2). In addition, *Eotrigonobalanus* is also part of the microfloristic assemblage (Plate I, K-L; Supplement S4). Pollen of the subfamily Engelhardioideae (Juglandaceae) is also abundant (min:

0.4 %, max: 21.7 %) and shows a decreasing trend towards Sequence m6 (20.9 – 20.7 Ma) (Fig. 2). Genera such as *Liquidambar*, *Elaeagnus* (Supplement S4), *Acer*, *Fagus*, *Nyssa* or *Artemisia* are sporadically present as are spores of pteridophytes. The most abundant palynomorphs are the gymnosperms *Pinus* (0.4 % to 32.7 %) and *Cathaya* (0 % to 27.1 %). Pollen grains of *Pinus* subg. *Strobus* outnumbered *Pinus* subg. *Pinus* in all samples (Fig. 2). The subdivision of *Pinus* at subgeneric level and of bisaccate pollen in general was often hindered by limited preservation. The sum of bisaccate

representatives fluctuated through the entire analysed time interval and reached the highest relative abundance of 120.8 % at 533.45 mbsf (Fig. 3). The Shannon-Wiener Index (H(*s*)), generally varied between 1 and 2 indicating relative low diversity, but shows distinct peaks >2 between 560.01-572.15 mbsf, 529.97-533.54 mbsf and 495.72-495.91 mbsf (Fig. 3).

### 3.3 Estimated palaeoclimate of NJSS

The bioclimatic analysis indicates humid warm-temperate climatic conditions for the entire dataset (Fig. 4). Average estimated

MAT for the entire record was 14.6 °C ± 4.0 °C. Palaeoclimatic estimates of CMMT are 5.2°C ± 5.3 °C and of WMMT 24.1°C ± 2.9 °C in average. All average palaeoclimatic estimations of each individual sequence are summarized in Table 1. The palaeoclimatic conditions of the New Jersey hinterland remained relatively constant over the preserved interval. Considering each investigated sequence individually, an average weak cooling and warming is reflected within the humid warm-temperate climate. Around the OMB (Fig. 4) between ~ 533 mbsf and ~ 529 mbsf a stepwise decline to lower temperatures is reflected

(MAT and WMMT ~ -3°C; CMMT ~ -5°C). The only estimated palaeoclimatic parameter which remained fairly constant over the analysed time is MAP (Fig. 4), reflecting humid conditions (always >1000mm).

## 4. Discussion

### 4.1 Taphonomy of terrestrial palynomorphs

Deposition of terrestrial palynomorphs in marine sediments depends on multiple factors (e.g. pollen production, transport

mechanisms, shoreline distance; van der Kaars, 2001). Studies concerning deposition of pollen and spores in neritic marine



sediments (Mudie and McCarthy, 1994; van der Kaars, 2001) indicate a reliable comparability of the microfloristic assemblage in marine sediments with the contemporary onshore vegetation. Because the palaeo-shelf break only transgressed the Site M0027 during the Early Miocene (McCarthy et al., 2013) dominance of bisaccate grains is expected, reflecting their preferential transport in wind and water (McCarthy et al., 2003).

Today the most important transport mechanism of pollen and spores into the marine realm along the North American coast are the westerlies (Mudie and McCarthy, 1994). Prevailing westerly winds were probably already established since the middle Oligocene. A further, but subordinate transport mechanism is pollen input via rivers and streams into the NJSS. Fluvial influence has been identified based on the sedimentological record (Miller et al., 2013a) for the NJSS. Downslope mass transport is not expected to be a major factor at Site M0027 prior to the transgression of the shelf break within sedimentary

Unit 6/sequence m5.8 (Fig. S1), except during glacioeustatic lowstands – then the ratio of terrestrial vs. marine palynomorphs ("P:D") helps identify resedimentation (McCarthy et al., 2013).

**4.2 Palaeoforest composition of the New Jersey hinterland**

Our results indicate that the Oligocene hinterland of the NJSS was covered by dense forests, with mesophytic forest growing on dry soils being the most widespread and diverse forest type on areas of low relief. Topographic higher elevations, and/or

drier areas were probably inhabited by conifers such as Pinaceae. Oaks (*Quercus* spp.) are the main element dominating the mesophytic forest of eastern North America in the past and at present (Abrams, 1992). Today ~90 different *Quercus* species are found in North America of which ~35 species are present in eastern North America (eFloras, 2008). Quercus probably appeared during the early Eocene (Grímsson et al., 2016 and references therein).

A late Eocene to early Oligocene palynofloristic record from southern Mississippi and Alabama (Oboh and Reeves Morris,

1994; Oboh et al., 1996) revealed that oaks became dominant at the Eocene-Oligocene boundary in the southeastern USA, probably indicating a vegetation shift triggered by cooling, as already suggested by Frederiksen (1991). In North America, the fossil record of *Quercus* implies that four infrageneric groups had existed during the Cenozoic (Denk et al., 2010 and references therein). Three of them persist in the modern North American vegetation, and two of these (Group Lobatae and Quercus) exist east of the Cordillera (eFloras, 2008). The analysed *Quercus* pollen grain sculptures from our study have affinities with Group

Quercus (white oaks; Plate I, C-D), Quercus/ Lobatae (white/ red oaks; Plate I, E-H) and aff. Group Protobalanus (golden-cup oaks; Plate I, I-J). These findings suggest the presence of a diverse *Quercus* population at subgeneric level during the Oligocene. Alternatively, the pollen ornamentation of Group Protobalanus may also indicate an ancestral lineage of the Protobalanus-Quercus-Lobatae clade (Grímsson et al., 2015).

The NJSS microfloristic assemblage also suggests the presence of *Eotrigonobalanus* (Plate I, K- L), an extinct Fagaceae

lineage. *Eotrigonobalanus* was a typical member of evergreen forests of the Palaeogene and flourished in different ecological habitats (Walther, 2000). The fossil record of *Eotrigonbalanus* in North America is very fragmentary (Grímsson et al., 2016). In Europe, Eotrigonobalanus was an accessory element within the evergreen forest and became dominant during the Late Eocene (Walther, 2000). In situ pollen grains from Texas referred to as *Amentoplexipollenites* (late Rupelian to Chattian;





Crepet and Nixon, 1989) are very similar with those of our investigation as well as with *Eotrigonobalanus* pollen grains of Central Europe (Denk et al., 2012). After Grímsson et al. (2016) a main characteristic difference between *Amentoplexipollenites* (Crepet and Nixon, 1989) and pollen grains of *Eotrigonobalanus* from British Columbia (Eocene) and Wyoming (Cretaceous) is the much thinner nexine of the latter. Evidence for the existence of *Triogobalanopsis*, another extinct

Fagaceae lineage in North America, is known from British Columbia (Eocene; Grímsson et al., 2016). All analysed pollen grains encountered in this study belonged to *Eotrigonobalanus*, however, we cannot rule out that this extinct lineage flourished in the New Jersey hinterland too.

Oaks are relatively persistent floristic elements within the mesophytic forest when comparing the Oligocene and late Mid-Miocene (Prader et al., 2017); however, significant floristic shifts within the mesophytic forest occurred. Relative abundances

of pollen grains of Engelhardioideae or *Carya* in our investigation reveal similar values when compared to data from the eastern coast of the Gulf of Mexico and South Carolina for the Late Eocene/Early Oligocene (Frederiksen, 1991; Oboh and Reeves Morris, 1994; Oboh et al., 1996). The recorded pollen percentages differ significantly from the values revealed for the late Mid-Miocene (Prader et al., 2017).

The Engelhardioideae (Juglandaceae) might have had a greater ecological range in the Oligo-Miocene than reflected by their

living representatives (Kvaček, 2007). However, the decreasing relative abundance of Engelhardioideae pollen in the long-term trend from the middle Oligocene to the late Miocene reveals that this group was not as competitive as other taxa concerning environmental changes. Accordingly, this group is not prominent any more in the pollen record of the late Mid-Miocene (Prader et al., 2017). If the drop of relative abundances of the Engelhardioideae was temperature-dependent, subsequent cooling events after the Mi-1 inception were probably necessary to induce the disappearance of the taxon (Prader

et al., 2017).

In comparison, other tree taxa like *Fagus* or *Liquidambar* were more constant elements of the mesophytic forest of the Miocene (Prader et al., 2017) than during the Oligocene. The fossil records of Fagus (Denk and Grimm, 2009) and Liquidambar (Manchester, 1999) for the Oligocene are fragmentary. These genera were also not prevalent in the New Jersey hinterland during the analysed time interval.

Fagus appeared the first time in the fossil record during the late early Eocene in western North America (Manchester and Dillhoff, 2005) and was continuously widespread during the late Oligocene in the western part of the continent and Europe (Kvaček and Walther, 1991). Its radiation and diversification led to a first occurrence peak in the Miocene (Denk and Grimm 2009). Our investigations indicate that *Fagus* was only a minor vegetation element in the hinterland of the NJSS during the early Miocene, but was a persistent component during the Mid-Miocene (Prader et al., 2017). Like today, the east coast of

North America was never a centre of biodiversity of beeches.

Contrary to *Fagus* and its spatiotemporal distribution, the Atlantic east coast is currently a hot spot of biodiversity of the genus *Carya* (Wen, 1999). In our record, the relative occurrences of this genus were as low as those of Fagus. *Carya* first appeared in the Palaeocene in North America (Manchester, 1987) and the first radiation began in the early Miocene (Zhang et al., 2013). Zhang et al. (2013) suggest that the Appalachian uplift phases created new habitats, which led to a diversification of the genus.



This might explain the increase of the pollen grain counts of *Carya* in the Mid-Miocene where *Carya* became the prevalent genus of the Juglandaceae (Prader et al., 2017).

**4.3 Terrestrial ecosystem responses to glacial events of the Mid-Oligocene to the Early Miocene.**

The entire Oligocene epoch is characterized by periodic climate oscillations associated with orbital insolation changes (Pälike
et al., 2006). These orbital climate changes triggered the built-up and decay of Antarctic Ice Sheets, leading to substantial glacioeustatic sea level oscillations (Pekar et al., 2002, 2006; Wade and Pälike, 2004, Pälike et al., 2006).
Reconstructions of bottom water temperatures in the equatorial Pacific (Mg/Ca temperature records of Site 1218) indicate low temperature variations for the Oligocene (on average 3.7 °C ± 1.5 °C; Lear et al., 2004). However, rapid cooling of 2 °C is associated with isotope events Oi-2b and Mi-1 (Lear et al., 2004). Although the limited temporal resolution of our record from
the NJSS inhibits the evaluation of vegetation responses to periodic orbital changes, it may be used to test the potential impacts of particularly strong Oligocene climate shifts. Specifically, fluctuations of the relative abundances of vegetation units can be interpreted in terms of possible movement signals.
However, the indications of such signals within the terrestrial palynomorph signature are weak. Possibly distinct marine influence hampered the regional impact by moderating the global climate changes.
Our reconstructed palaeoclimate shows humid-warm temperate conditions under which only little altitudinal shifts took place. One example for a altitudinal movement signal could be the first peak of the meso- and microthermal Pinaceae (Fig. 4) at 29.3 -29.0 Ma. The most plausible explanation for this peak might be the impact of the unnamed $\delta^{18}$O excursion at ~ 29.1 in the records from New Jersey (Pekar et al., 2002) and from the tropical Pacific (Site 1218: Wade and Pälike, 2004). However, relative pollen abundances of Vegetation Unit 4 remained relative stable (Fig. 3), indicating that either the amplitude of climatic
factors (e.g. cooling) was too small to induce a strong reduction of the mesophytic forest or the regional vegetation units had a high degree of resilience to the climate forcing.
The first noteworthy reduction of Vegetation Unit 4 and a simultaneous increase of bisaccate Pinaceae probably coincide with the Oi2-a event at ~28.3 Ma (Pekar et al., 2002) and/or the ~28.5-Ma-$\delta^{18}$O increase (Pekar et al., 2002). The conifers spread into the lowland and partly replaced Vegetation Unit 4 (Fig. 3). The idea of downward movement of the conifers is reinforced
by the negatively correlating P:D ratio, which indicates an increasing shoreline vicinity and thus only a low chance of transport-induced increase in bisaccate pollen percentages.
Differing to the placing of the Mi-1 event at 23.03 Ma based on orbital tuning (Liebrand et al., 2011), it was placed at 23.8 Ma at the New Jersey margin by Pekar et al. (2002), who postulate an apparent sea-level drop of 56 m connected to the Mi-1 event. A definitive placement of the Mi-1 event is however difficult for Site M0027, since it probably corresponded with a strong
sea-level lowstand creating the sequence boundary of Sequence O6 (Browning et al., 2013). The increased input of bisaccate pollen at the beginning of Sequence O6 (probably around the OMB) is consistent with findings of Kotthoff et al. (2014) which indicate that abiotic factors ameliorated the expansion of conifers (Vegetation Units 1 and 2) and led to a relative decrease of Vegetation Unit 4 (mesophytic forest, Fig. 3).



A temperature drop might be visible within all parameters at ~530 mbsf in our data, with a stepwise MAT-decline of ~3°C. It probably matches with the MAT-drop of ~4°C described by Kotthoff et al. (2014) at the same depth. A coeval increase of Vegetation Unit 7 (coastal), probably reflecting dryer and cooler conditions, is also consistent with Kotthoff et al. (2014). However, MAP does not decrease around this time interval, the dataset rather suggests an average overall stable trend in precipitation suggesting more occupation of coastal plain. A further decline of Unit 4 occurred within Sequence m6 (~20.9-20.7 Ma; Fig. 3), probably reflecting a late Aquitanian vegetation change provoked by the unnamed $\delta^{18}O$ increase at ~20.8 Ma (Browning et al., 2013).

The generally fluctuating patterns of Vegetation Units 3 and 4 and of the (H(s))-Index represent indirect confirmation of climate change in the hinterland of the NJSS in phase with depositional changes of the Oligocene and early Miocene sediment succession at Site M0027. Peaks of Vegetation Units 5 and 3 coincide with all regression phases (glacial phases) discussed above, which exposed shallow shelf areas allowing for the spreading of substrate-depending forest formations (Fig. 3). The simultaneous expansion of Vegetation Units 3 and 5 percentages and concomitant increase of the (H(s))-Index indicate a quite diverse low elevation forest and thus a decreasing distance to the shoreline, facilitating pollen grain input during regression phases and/or a larger altitudinal gradient.

## 5. Conclusion

The comparison of recorded pollen percentages of this investigation with the late Mid-Miocene values (Prader et al., 2017), revealed a contrasted spatiotemporal distribution of mesic taxa in the New Jersey hinterland, probably caused by changing environmental conditions which in the long-term trend led to an enhanced floristic turnover. The presence of different pollen grain sculptures within the genus *Quercus* suggests a diverse *Quercus* population at subgeneric level. The relatively high abundances of *Quercus* pollen grains emphasise the dominance and significance of this taxon within the mesophytic forest. The sporadic occurrence of the extinct Fagaceae genus *Eotrigonobalanus* in the New Jersey hinterland widens the understanding of its distribution pattern in the Northern Hemisphere during the Cenozoic. The bioclimatic analysis reveals a relative climatic stability during the middle Oligocene to Early Miocene reflecting within humid warm temperate condition. However, a stepwise weak cooling signal during this interval is reflected in calculated MAT, WMMT and CMMT and might correspond to the Mi-1 event (MAT and WMMT ~ -3°C; CMMT ~ -5°C).

The limited temporal resolution of our pollen record from the NJSS inhibited the evaluation of vegetation responses to periodic orbital changes of the Oligocene and Early Miocene. Nevertheless, the terrestrial palynomorph signature demonstrate weak fluctuations signals through these time intervals, which allow the interpretation in terms of possible movement signals caused by periodic climate oscillations. The observed movement signals of different intensions are probably best reflected within peaks of meso- and microthermal Pinaceae, probably corresponding to the unnamed $\delta^{18}O$ excursion at ~ 29.1, the Oi2-a event at ~28.3 Ma and/or the ~28.5-Ma-$\delta^{18}O$ increase and to the Mi-1 event at ~ 23.8 Ma. Regression phases (glacial phases) exposed



shallow shelf areas and allowed the spreading of substrate-depending forest formations. The spreading of Vegetation Units 3 and 4 during glacial phases represented thus an indirect confirmation of climate change.

### 6. Acknowledgements

We thank the entire Integrated Ocean Drilling Program (IODP) Expedition 313 Scientific Party and the staff of the IODP Core Repository Bremen. We acknowledge A. Krueger, Palynology Lab at Brock University; E. Czymoch and K. A. Harps, University of Hamburg, for sample processing and Y. Milker, M. Theodor and R. Walter, for support at the SEM and J. Reolid for sea level oscillations discussion. The research was supported by the German Science Foundation (DFG; Project Ko 3944/5). DRG's participation was further supported by a grant from the Natural Sciences and Engineering Research Council of Canada (RGPIN 2016-04337).

### 7. Author contributions

Author contributions: U.K. and F.M.G.M had the research idea, S.P. wrote the MS with substantial contributions of all authors. S.P. counted and analysed data with support of T.D. (pollen identification), G.S. (statistical analyses) and D.R.G. (bioclimatic analyses).

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

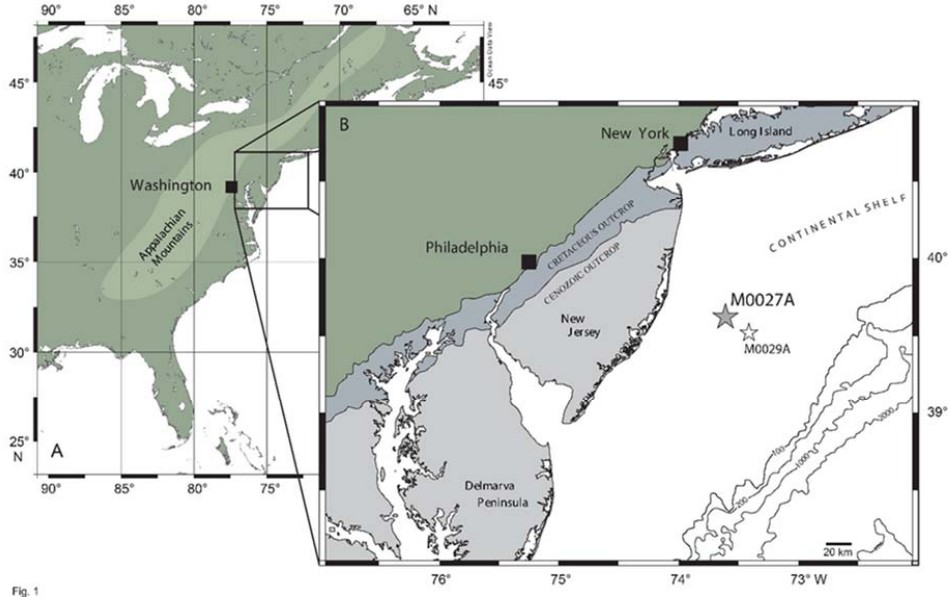

Figure 1. A. Location of the study area in eastern North America. B. Detailed map of the New Jersey area. Grey star indicates position of Site M0027, white star: position of Site M0029. After Mountain et al. (2010), Schlitzer (2011), Kotthoff et al. (2014).




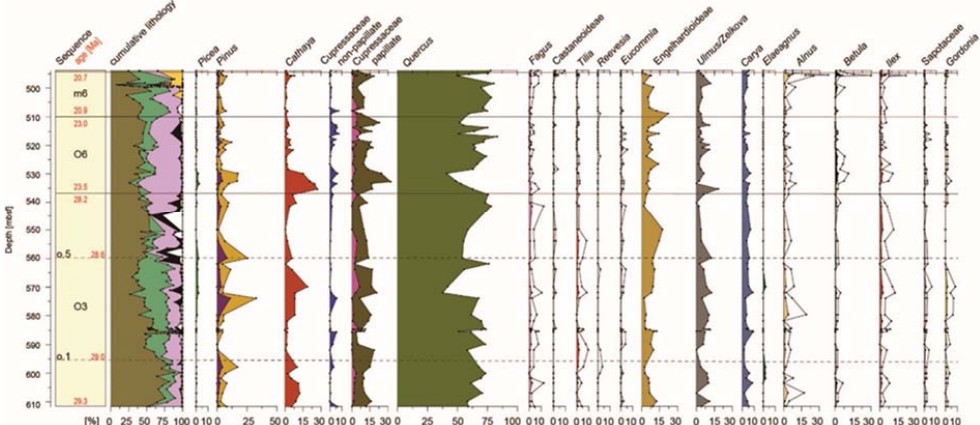

**Figure 2.** Relative abundances of selected pollen grain taxa for Site M0027A plotted against depth (pollen sum excluding bisaccates). Transparent areas in plots of *Fagus*, *Tilia*, *Betula*, *Alnus*, *Ilex*, Sapotaceae, *Reevesia*, *Elaeagnus*, *Eucommia*, *Gordonia* denote 5x exaggerated values. *Pinus* subg. *Strobus*: violet; *Pinus* subg. *Pinus* and unidentifiable *Pinus* pollen grains: yellow. Cupressaceae non-papillate type: blue; Cupressaceae presumably with papilla: pink; Cupressaceae with a papilla: brown. Sequence boundaries and age model after Browning et al. (2013) and Miller et al. (2013a, 2013b) with dashed lines indicating intrasequences. Cumulative lithology after Miller et al. (2013a): brown: clay and silt; green: glauconite; violet: quartz sand; yellow: medium and coarser quartz sand; black: carbonate; white: mica and other.

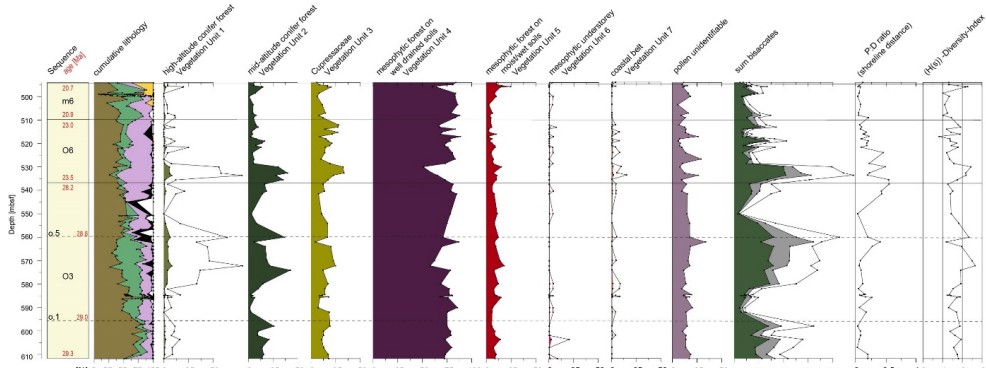

**Figure 3.** Vegetation units of the NJSS hinterland during the middle Oligocene to Early Miocene plotted against depth. Taxa are assigned as listed in Table S3 (transparent: 10x exaggerated: Vegetation Unit 1, Vegetation Unit 6, Vegetation Unit 7; sum of bisaccates: grey: unassigned bisaccate pollen grains; green: percentages of destroyed parts of bisaccate pollen grains (H(s))-Diversity-Index. Pollen-dinoflagellate ratio (P:D) indicating site-shoreline distance. Sequence boundaries and age model after Browning et al. (2013) and Miller et al. (2013a, 2013b): dashed lines: intrasequences. Cumulative lithology after Miller at al. (2013a): brown: clay and silt; green: glauconite; violet: quartz sand; yellow: medium and coarser quartz sand; black: carbonate; white: mica and other.





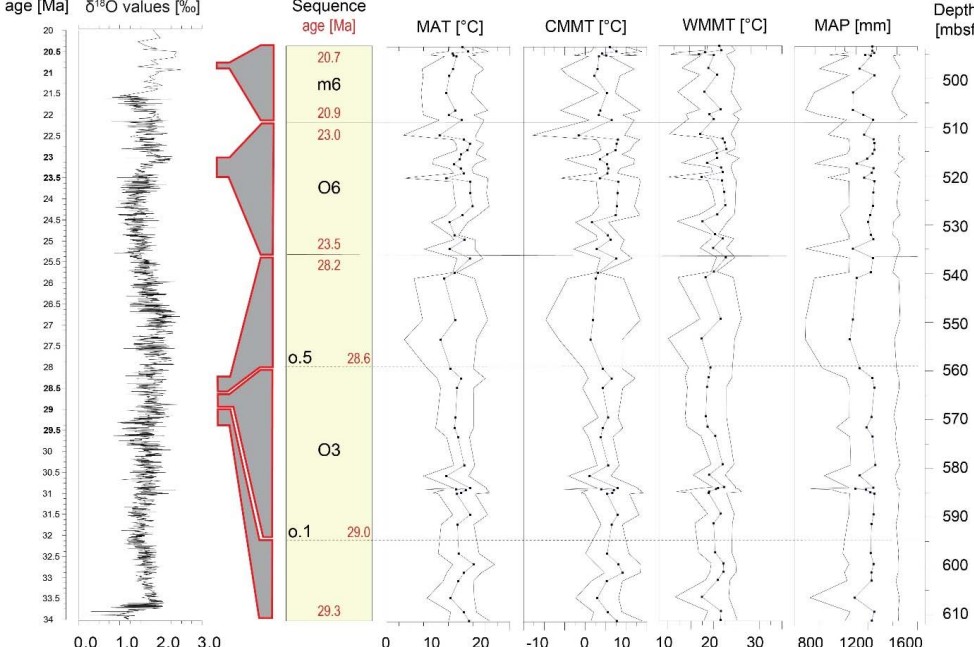

**Figure 4. Calculated palaeoclimate during the middle Oligocene and early Miocene for the hinterland of the NJSS based on identified pollen grains from Site M0027 following the bioclimatic analysis after Greenwood et al. (2005) and Prebble et al. (2017) plotted against depth. Compared with δ¹⁸O values of ODP Site 1218 (equatorial Pacific) after Wade and Pälike (2004) plotted against age. Please consider the different age models. MAT: mean annual temperature; CMMT: coldest moth mean temperature; WMMT: warmest month mean temperature and MAP: mean annual precipitation; outer lines: error of estimated values (present day climate parameters of Toms River (New Jersey): MAT 11.7 C°; CMMT: 5.7 C°; WMMT: 17.7 C°; MAP: 1239 mm, after US Climate Data, 2017)**





**Plate I SEM images of selected pollen grains of Site M0027A (574.05 mbsf). Overview (A, C, E, G, I, K, M, N O) and detail (B, D, F, H, J, L, P). A-B:** *Fagus*; **C-D:** *Quercus* **Group Quercus; E-H:** *Quercus* **Group Quercus/ Lobatae; I-J:** *Quercus* **aff. Group Protobalanus; K-L:** *Eotrigonobalanus*; **M-N:** *Gordonia*; **O-P:** *Tilia*. **Scale bar 10µm (A, C, E, G, I, K, M, N O), 1µm (B, D, F, H, J, L, P).**

| Sequence | MAT [°C] | CMMT [°C] | WMMT [°C] | MAP [mm] |
|----------|----------|-----------|-----------|----------|
| O3 | 14.0 (±4.1) | 4.1 (±5.1) | 23.9 (±3.0) | 1312.2 (±293.9) |
| O6 | 15.0 (±3.6) | 5.5 (±5.2) | 24.5 (±2.4) | 1332.6 (±278.8) |
| m6 | 13.8 (±4.8) | 4.4. (±6.0) | 23.5 (±3.6) | 1297.4(±324.2) |
| m5.8 | 15.4 (±2.5) | 6.1 (±4.0) | 24.8 (±1.9) | 1359.9 (±270.1) |

**Table1. Summary of average climatic values and average errors of estimated values for each analysed sequence from the middle and late Oligocene and early Miocene (Sequence m5.8 compromise only one analysed sample).**