# Peer review of "Plants in movement – Floristic and climatic characterization of the New Jersey hinterland during the Palaeogene-Neogene transition in relation to major glaciation events"

_Biogeosciences, 2017_

## Referee Comment (RC1) · Anonymous Referee #1 · 1 Feb 2018

This paper shows pollen data from the late Oligocene- early Miocene part of core site M0027 (IODP expedition 313) taken from the offshore of New Jersey. This subject could have potentially been interesting as there is not much information about vegetation dynamics over that time period, including one significant glaciation Mi-1 - marking the Oligocene-Miocene transition. Nevertheless, in my opinion there are significant limitations and flaws in the study that precludes its publication in high-impact reviews such as Biogeosciences.

1. The sedimentary record is very discontinuous. There are big "jumps" in time - in

some cases of many million years, for example with no sedimentation between 28.2-23.5 Ma or between 23-20.9 Ma - and this limits the pollen data and their interpretation in terms of vegetation/climate dynamics. The authors say at some point that the glacial (sea-level lowstand) part of the sequences are missing. However, the diagrams show a slice of the sequence that would cover one of the most important glaciations at that time - the Mi-1 glaciation at around 23 Ma - but there is barely any discussion about it. Are there any vegetation patterns that would show this significant environmental cooling?

In this respect, the pollen diagrams are very difficult to understand because they are represented continuously even though there are significant hiatuses and graphically one tries to see cycles that are obviously not there.

2. The discussion is very short, shallow and naïve. The adjective "weak" is used twice in the abstract describing the vegetation and paleoclimatic estimations but no further information is given describing the patterns, which seem to be shown in Figs 2 or 3 (i.e., peaks in conifers). I feel the authors did not squeeze the data enough to obtain conclusion. Do you really believe climate was "a straight line" between 30 and 20 Ma??? For example, the authors first say that the hiatuses "inhibits the evaluation of vegetation responses to periodic orbital changes". However, some of the sections studied are about 500,000 years, long enough to at least record orbital eccentricity changes. At the beginning of section 4.3 the authors say that the terrestrial pollen signal could be hampered in the marine environment but they do not explain how.

3. There are some pollen changes in some of the studied sections such as O3 or O6 with clear dynamics in the conifers that are barely discussed. However, some other events are discussed but are not very clear. For example, the authors in section 4.3 described a significant change, that they related with Oi2-a event at 28.3 Ma, characterized by an increase in Pinaceae. However, in Fig. 3 there seems to be a minimum in Pinaceae instead (sum-bissacates).

4. The vegetation is mostly described as "units" and this is very confusing as one has to be going back to the figures or, even worse, the supplementary document, where the taxa are described. Why not calling these units by their names? For example, high-altitude conifer forest, instead of unit 1?

5. The lithology plot shown in Fig. 3 is very confusing and very difficult to understand. Lithology of the studied core should be shown in standard lithological patterns so one can visualize if the pollen changes are at some point related with sedimentological, transport of taphonomical changes.

6. Eotrigonobalanus is described in the text as an extinct genus occurring in this sedimentary record. SEM seems to be necessary to classify this pollen taxon but no discussion is found in the manuscript concerning how abundant it was or if it occurred continuously throughout the record.

7. There are some plant taxa that show affinities with subtropical climates such as Engelhardia, Sapotaceae, Cupressaceae with papilla (probably Taxodioideae) that do not occur in the area at present probably due to climate cooling but one gets the impression that the climatic inferences made here are pretty much like at present.

---

## Referee Comment (RC2) · T. Denk (Referee) · 12 Feb 2018

The authors investigated a palynological sequence from eastern North America spanning the Oligocene Miocene boundary in order to infer climate evolution over this time interval, possible movements of vegetation units along an altitudinal gradient, and to compare this with major glaciation events during this time. Climate reconstruction uses a nearest living relative approach and altitudinal vegetational shifts are investigated by the relative contribution of so-called "artificial vegetation units" to the entire palyno-assemblage. Some pollen taxa are documented using LM and SEM.

This paper is not ready for publication. There are problems with presentation of the data (inappropriate pollen diagram) and general problems (coding of vegetation units). However, if the authors are willing to undertake a full revision I do see a potential for this paper to become a valuable contribution to Biogeosciences. I start my report with a few general comments followed by specific remarks.

General comment 1: I think you should provide a full pollen diagram. The present way of presenting the pollen data is not acceptable, especially since you want to trace possible correlations with climate change. It may be that rare exotic elements disappear from the pollen record at some point, but this is not seen in your very cryptic diagram. Secondly you must indicate hiatuses in Figures 2–4. It looks very odd as it is.

In this context I also wonder how you distinguish Larix from Pseudotsuga, and Tsuga canadensis from T. caroliniana. Please explain. Further, I wonder about the presence of Cedrus in E North America. This finding would need to be verified using SEM.

General comment 2: In Supplement S3 plant taxa are assigned to vegetation units and this assignment has implications for inferring altitudinal shifts of vegetation units (in response to cooling/warming). To my knowledge, it is impossible to score each taxon for one particular vegetation unit. Just taking Flora of North America it is clear that many of the reported taxa have wide ecological ranges and occur in more than one vegetation unit: A few examples are Pinus (2, 5 and 7), Apiaceae (4 and 7), Artemisia (1, 2, 6, 7), Ericaceae (1-6), Fabaceae (1-7), Hamamelidaceae (4, 5, 7) etc. I think this must be corrected.

I also wondered about the vegetation unit "Cupressaceae". These may include almost everything, from swamp forest to sand dunes, to mixed mesophytic forest, to rocky cliffs etc.

General comment 3: relates to section 3.3, line 25 It would be very illuminating to know which changes in taxon composition/pollen frequencies accounted for this purported drop in temperature. Also for the section further down on climate fluctuations, it would

be helpful to name the taxa that account for changes in temperature parameters.

General comment 4: relates to section 4.2, lines 15, 16 Even if just taking extant pines of E North America a great number of different ecologies are encountered: Forest, sandy soils, sand dunes, bogs, and typically occurring in flatwoods. The latter would correspond to your vegetation unit 5, I think, sand dunes to VU 7 etc. The same is true for oaks, especially if you have sections Quercus and Lobatae both well-drained and wet soils forests are equally possible environments. You may want to update to the current classification of Quercus (Denk et al. 2017).

General comment 5: page 7, line 29 Something is wrong here. The modern distribution of Fagus grandifolia is just exactly in E North America. Not precisely on the coast but close to it. Check in GBIF. See also Bennet 1985, J. Biogeography. I would expect fairly high percentages of Fagus pollen in a modern pollen diagram. Was Fagus less common during some of the time intervals investigated by you?

General comment 6: relates to section 4.3 I must admit that I had hard times reading this part. p. 8, line 16: Which Pinaceae? On the pollen diagram I see Pinus and Cathaya – Cathaya is more like a mesophytic element and fits well into vegetation unit 4. Also consider that "several conifer taxa may also have been part of hammocks within peat-forming vegetation (e.g. Cathaya, Sequoia, Taiwania) and of raised bogs (Sciadopitys; Schneider, 1992; Dolezych & Schneider, 2007)." [from Denk 2016]. All in all, I am not convinced by the claimed correlation between glaciations and fluctuations in the palynological record investigated for this study. Again, I think the entire pollen profile must be presented, and if the authors want to make a case for shifts in some Pinaceae taxa reflecting cooling or warming then this should be illustrated based on a detailed pollen diagram.

General comment 7: Aspects of the paper that are insufficiently addressed in the current version Biogeographic aspects could be discussed when making use of the full pollen diagram. For example, I noted that Eotrigonobalanus, "Eleagnus", and Cedrus

provide links to Europe. Eucommia, Cathaya, Sciadopitys provide geographic links with East Asia but were widespread in the N Hemisphere during the Cenozoic and . Cedrelospermum and Eotrigonobalanus are extinct taxa. Are these taxa represented in all time slices covered by this study, is there a pattern of extinction? How does the pollen assemblage studied here compare to the Brandon Lignite?

I also noted that the Fagus pollen you figure is very interesting: it has a long and narrow colpus reaching almost to the poles of the grain. This is typical of subgenus Engleriana and the most distinct species Fagus grandifolia within subgenus Fagus.

Specific comments: Throughout the text: It is early Eocene, middle Miocene, etc. not Early Eocene, Middle Miocene

Page 1, line 15. "altitudinal spatial and long-term temporal vegetation migration" Very much information in one sentence. I don't understand what you mean with "altitudinal spatial" Are you assigning taxa to vertical vegetation zones? Same with "long-term temporal". Perhaps better to just say long-term.

Page 1, line 18. "To infer possible topographic palaeovegetation movements" Does this mean the same as "altitudinal spatial" above. This is very confusing, please re-phrase.

Page 1, line 23. "Biotic responds to environment change" change to: Biotic responses to environmental change.

Page 2, line 9. 20 or 23?

Page 7, line 6. "rule out that this extinct lineage" change to: rule out that Trigonobalanopsis

Page 7, line 14. "had a greater ecological range" change to: had a wider ecological range

Page 7, line 29. "persistent" do you mean "common"?

Page 7, line 31. "Contrary to Fagus and its spatiotemporal distribution, the Atlantic east

coast is currently a hot spot. . ." Please re-phrase.

Page 8, line 1. "where Carya became the prevalent genus" Please re-phrase.

Page 8, line 13. Do you want to say that the maritime setting of your sample sites buffered possible regional climate change. And that this could explain the weak signal in the palynological record?

Page 8, lines 27, 28. Please re-phrase.

Page 9, line 1. "might" or: is?

Page 9, lines 17, 18. "a contrasted spatiotemporal distribution ..." This does not make sense, please re-phrase. What exactly do you want to say? Same for "enhanced floral turnover". Please re-phrase.

Page 9, lines 25 ff. Please re-phrase. And re-think the possible movements.

Figure 4. Oxygen isotope curve. You may want to colour warming and cooling trends using red and blue. This would make it easier to read the figure.

Supplement, Plate S4-ii. "Eleagnus" looks similar to Boehlensipollis hohli from Rupelian to Aquitanian strata of France, Belgium, and Poland (Sittler et al., 1975; Stuchlik et al., 2014). Affinities are possibly with Eleagnus but possibly also with other genera. Still, you have a nice example of a European-E North American disjunction here.

References: Bennet, K. D. 1985. The spread of Fagus grandifolia across eastern North America during the last 18 000 years. Journal of Biogeography, 12: 147-164.

Denk, T. 2016. Palaeoecological interpretation of the late Miocene landscapes and vegetation of northern Greece: a comment to Merceron et al., 2016 (Geobios, doi:10.1016/j.geobios.2016.01.004). Geobios, 49: 135-146.

Denk, T., Grimm, G. W., Manos, P. S., Deng, M. & Hipp, A. 2017. An updated infrageneric classification of the oaks: review of previous taxonomic schemes and synthesis of evolutionary patterns. In: Gil-Peregrin, E., Peguero-Pina, J.J., Sancho-Knapik, D. (eds) Oaks Physiological Ecology. Exploring the Functional Diversity of Genus Quercus. Tree Physiology 7, pp. 13-38. Springer Nature, Cham, Switzerland.

Sittler, C. et al. 1975. Extension stratigraphique, répartition géographique et écologie de deux genres polliniques paléogènes observés en Europe occidentale: Aglaoreidia et Boehlensipollis. Bulletin de la Société Botanique de France, 122, supp. 1: 231-245.

Stuchlik L, Ziembińska-Tworzydło M, Kohlman-Adamska A, Grabowska I, Słodkowska B, Worobiec E, Durska E. 2014. Atlas of pollen and spores of the Polish Neogene. Volume 4. Angiosperms (2). Kraków: W. Szafer Institute of Botany, Polish Academy of Science.

---

## Author Comment (AC1) · 16 Apr 2018

Answer to referee 1.

We can follow several comments made by Referee 1 and will change the manuscript accordingly. In some other case, the comments have a rather general character, e.g. relating to weaknesses of our record regarding the hiatuses between the different sequences. We are very much aware of the discontinuous nature of the presented record, which is typical of palynological records from shelf regions, such as the mentioned hiatuses between sequences or concerning taphonomical aspects. This is offset by the importance of making direct (single core) land-sea correlation since the work is part of an IODP site with many parallel analyses, sampling a large integrated catchment area, and the absolute scarcity of well-dated palaeobotanical records for the eastern coast of North America so that our record, which comprises several relatively well-dated sequences, is in any case a very important addition which should be of interest for all studies dealing with the onset of the Miocene in the research area.

1. The sedimentary record is very discontinuous. There are big "jumps" in time – in some cases of many million years, for example with no sedimentation between 28.2-23.5 Ma or between 23-20.9 Ma - and this limits the pollen data and their interpretation in terms of vegetation/climate dynamics. The authors say at some point that the glacial (sea-level lowstand) part of the sequences are missing. However, the diagrams show a slice of the sequence that would cover one of the most important glaciations at that time - the Mi-1 glaciation at around 23 Ma - but there is barely any discussion about it. Are there any vegetation patterns that would show this significant environmental cooling? In this respect, the pollen diagrams are very difficult to understand because they are represented continuously even though there are significant hiatuses and graphically one tries to see cycles that are obviously not there.

To facilitate the understanding of the hiatuses in the analysed time interval, we plan to graphically indicate these hiatuses in the diagram. Compare also comments by referee 2. We can follow Referee 1's argument that the discussion about the Mi-1 event is relatively short. However, as mentioned in the discussion the placement of the Mi-1 event at the New Jersey margin is not clear and since today, no clear conclusions exist if the Mi-1 event is expressed at the New Jersey margin or not. We suggest that we discuss the temporal placement of the Mi-1 event at the New Jersey margin in more detail, also in relation to the increased input of bisaccates pollen grains during cold events. 2. The discussion is very short, shallow and naïve. The adjective "weak" is used twice in the abstract describing the vegetation and paleoclimatic estimations but

no further information is given describing the patterns, which seem to be shown in Figs 2 or 3 (i.e., peaks in conifers). I feel the authors did not squeeze the data enough to obtain conclusion. Do you really believe climate was "a straight line" between 30 and 20 Ma??? For example, the authors first say that the hiatuses "inhibits the evaluation of vegetation responses to periodic orbital changes". However, some of the sections studied are about 500,000 years, long enough to at least record orbital eccentricity changes. At the beginning of section 4.3 the authors say that the terrestrial pollen signal could be hampered in the marine environment but they do not explain how.

We can only partly follow these comments. We assume that our record does not meet all expectations referee 1 has for a good climate record, and as discussed above, this is to some degree understandable, e.g. regarding the hiatuses. However, our study was in the first order designed to discuss vegetation movements during the time intervals reflected in our record and to contribute to palaeobiogeographical data. We do see a clear dynamic pattern in the New Jersey hinterland vegetation according to global climatic events where conventional palaeoclimatic reconstructions do not show similar fluctuations signals. This not only a problem in our record, but also in other publications using the nearest living relative approach or similar approaches to reconstruct climate variability. Thus, in our manuscript we want to point out that it is useful to think about new aspects within the development of approaches to quantify climatic dynamics in palaeorecords. We see our study as an example for such approaches. We did not intend to pretend that we have a continuing climate record without interruptions. The study was also not designed to detect orbitally-driven palaeovegetation changes. Indeed, some Sequences are about 500.000 years long and as well long enough to study orbital changes (actually we are preparing a manuscript in that direction focusing on another sequence from the same site), but such a study would have a different focus. What we wanted to state in section 4.3 is that the vicinity to the sea probably buffered the impact of different cooling events to some degree. We will rephrase the related sentences to make this clearer. 3. There are some pollen changes in some of the studied sections such as O3 or O6 with clear dynamics in the conifers that are

barely discussed. However, some other events are discussed but are not very clear. For example, the authors in section 4.3 described a significant change, that they related with Oi2-a event at 28.3 Ma, characterized by an increase in Pinaceae. However, in Fig. 3 there seems to be a minimum in Pinaceae instead (sum-bissacates).

We are going to include the conifer peaks, which show a clear dynamic pattern into the discussion in the context that, for this peaks in the literature no cooling events are described and could be connected to regional changes in the hinterland. Taking the uncertainty of our age model into account the huge input of bisaccate pollen grains around the intrasequence o.5 could be related to the 28. 5 Ma cooling event described in Pekar et al. (2002). We agree with the Referee 1 that Oi2-a event at 28.3 Ma is not really expressed, this point we are going to discuss in detail.

4. The vegetation is mostly described as "units" and this is very confusing as one has to be going back to the figures or, even worse, the supplementary document, where the taxa are described. Why not calling these units by their names? For example, high-altitude conifer forest, instead of unit 1?

We aimed at keeping the manuscript shorter by using the abbreviations, but we agree that it would facilitate to understand the text when the specific names of the units were used. We will change the text accordingly.

5. The lithology plot shown in Fig. 3 is very confusing and very difficult to understand. Lithology of the studied core should be shown in standard lithological patterns so one can visualize if the pollen changes are at some point related with sedimentological, transport of taphonomical changes. We understand this that way that referee 1 would prefer the typical signatures for clay, silt etc. and will change the figures accordingly.

6. Eotrigonobalanus is described in the text as an extinct genus occurring in this sedimentary record. SEM seems to be necessary to classify this pollen taxon but no discussion is found in the manuscript concerning how abundant it was or if it occurred continuously throughout the record.

We are going to show a diagram with the relative abundances of Eotrigonobalanus and other rare taxa, and the discussion about this genus will be a bit longer, including information on its abundancy in the pollen records. (Compare comments to T. Denk , referee 2.)

7. There are some plant taxa that show affinities with subtropical climates such as Engelhardia, Sapotaceae, Cupressaceae with papilla (probably Taxodioideae) that do not occur in the area at present probably due to climate cooling but one gets the impression that the climatic inferences made here are pretty much like at present.

The above mentioned genera are not distinguishable at lower systematic level. This topic has been discussed by several authors recently (e.g. Cupressaceae: Grimm et al. (2015) and Grimm and Potts (2016), compare also Prader et al. (2017)). We will discuss some aspects related to these taxa in more detail, for example the relative abundances of the Engelhardioideae of the late mid Miocene (Prader et al. 2017) compared to the Oligocene to early Miocene so that the reader could see that minor changes in the vegetation composition occurred in the New Jersey hinterland. It is however difficult to make comparisons with the present-day situation in case of Engelhardia. This aspect is also related to comments by referee 2 concerning the development of Fagus in the research area.

Literature cited: Grimm, G., Denk, T., Bouchal, J.M., Potts, A.J., 2015. Fables and foibles: a critical analysis of the Palaeoflora database and the Coexistence approach for palaeoclimate reconstruction. BioRxiv 016378; doi: http://dx.doi.org/10.1101/016378. Grimm, G.W., Potts, A.J., 2016. Fallacies and fantasies: the theoretical underpinnings of the Coexistence Approach for palaeoclimate reconstruction. Climate of the Past 12, 611-622. Pekar, S. F., Christie-Blick, N., Kominz, M. A. and Miller, K. G., 2002. Calibration 5 between eustatic estimates from backstrippingand oxygen isotopic records for the Oligocene. Geology 30, 903-906. Prader, S., Kotthoff, U., McCarthy, F.M.G., Schmiedl, G., Donders, T.H., Greenwood, D.R., 2017. Vegetation and climate development of the New Jersey hinterland during the late Middle Miocene (IODP Expedition 313 Site M0027). Palaeogeography, Palaeoclimatology, Palaeoecology 485, 854-868.

---

## Author Comment (AC2) · 16 Apr 2018

General comment 1: I think you should provide a full pollen diagram. The present way of presenting the pollen data is not acceptable, especially since you want to trace possible correlations with climate change. It may be that rare exotic elements disappear from the pollen record at some point, but this is not seen in your very cryptic diagram. Secondly you must indicate hiatuses in Figures 2–4. It looks very odd as it is. We are going to provide a full pollen diagram. We will graphically indicate the hiatuses between the different sequences in Figures 2-4 via gaps between the different sequences

and appropriate symbols. In this context I also wonder how you distinguish Larix from Pseudotsuga, and Tsuga canadensis from T. caroliniana. Please explain. Further, I wonder about the presence of Cedrus in E North America. This finding would need to be verified using SEM.

We agree with T. Denk that the differentiation between Larix and Pseudotsuga is not possible, thus we are going to combine Pseudotsuga and Larix (we will write Larix/Pseudolarix). Tsuga canadensis vs. T. caroliniana were distinguished via the presence or absence of echinae. We will discuss this differentiation in the text to make it clearer. Since we cannot be 100% sure that those grains we assigned to T. caroliniana cannot be other Tsuga pollen grains with echinae, we suggest that we use the neutral terms Tsuga sp. 1 and Tsuga sp. 2. For the climate reconstruction, we intend to use a combined climate data set of North American Tsuga spp.. Concerning Cedrus: Since we could not yet verify this finding via SEM, we will exclude this taxon from this publication. General comment 2: In Supplement S3 plant taxa are assigned to vegetation units and this assignment has implications for inferring altitudinal shifts of vegetation units (in response to cooling/warming). To my knowledge, it is impossible to score each taxon for one particular vegetation unit. Just taking Flora of North America it is clear that many of the reported taxa have wide ecological ranges and occur in more than one vegetation unit: A few examples are Pinus (2, 5 and 7), Apiaceae (4 and 7), Artemisia (1, 2, 6, 7), Ericaceae (1-6), Fabaceae (1-7), Hamamelidaceae (4, 5, 7) etc. I think this must be corrected.

We definitely agree with T. Denk here that many of the identified taxa have wider ecological optima this was the reason why we introduced the term "artificial" as we have mentioned in the text. The assignment of taxa to the different vegetation types goes back to Prader et al. (2017) and was similarly done by, e.g. Larsson et al. (2011). We have chosen the same system of assignment as in Prader et al. (2017) for a better comparability of both investigations. We suggest that we indicate the possible additional assignments of those taxa who can be assigned to more than one vegetation

unit in the supplements. Additionally, we would discuss in more detail why the taxa were assigned to the respective groups and to which unit they may also have been assigned.

I also wondered about the vegetation unit "Cupressaceae". These may include almost everything, from swamp forest to sand dunes, to mixed mesophytic forest, to rocky cliffs etc.

We agree, taxa within this family are highly variable in their ecological range. Due to the problematic identification at genus level this family is hard to handle (not only concerning climate reconstructions, but also concerning vegetation reconstructions). This was the main reason why this family was not assigned to a specific vegetation unit (see above!) and why it was excluded from the climatic reconstructions.

We will discuss this "unit" in higher detail and point, e.g. to the fact that macrofossils of Taxodium sp. are reported from the Miocene Brandywine flora (McCarthan et al., 1990), which would probably underline that the main input of Cupressaceae pollen grains with a papilla (during the Oligocene) probably derived from this taxon.

General comment 3 (relates to section 3.3, line 25): It would be very illuminating to know which changes in taxon composition/pollen frequencies accounted for this purported drop in temperature. Also for the section further down on climate fluctuations, it would be helpful to name the taxa that account for changes in temperature parameters.

In the first order the presence or absence of Gordonia pollen grains is responsible for the strongest fluctuations in the estimated bioclimatic values. The stepwise drop in MAT around the OMB is as well caused by Gordonia (lacking in the lowest estimated value around the OMB). We are going to include this information in the text.

General comment 4: relates to section 4.2, lines 15, 16 Even if just taking extant pines of E North America a great number of different ecologies are encountered: Forest, sandy soils, sand dunes, bogs, and typically occurring in flatwoods. The latter would

correspond to your vegetation unit 5, I think, sand dunes to VU 7 etc. We agree that the ecological range of Pinus is much greater and that it occurs in more than one vegetation unit (see above). Considering that Pinus produces particularly high amounts of pollen, but that the percentages of the Pinus are still not particularly high in our record, we still assume that this taxon in our record is rather part of the highlands. If Pinus was also a frequent taxon within the lowland or even coastal vegetation, we would expect a higher input of this pollen grain type into the core sediments (compare Hooghiemstra et al., 2006; Kotthoff et al., 2008, who discuss records with Pinus dominating coastal vegetation during certain intervals). In addition, as discussed above, we also follow an approach which was already used for other datasets and which is used in our manuscript to allow a direct comparison. As suggested above, we will discuss the assignment in more detail.

The same is true for oaks, especially if you have sections Quercus and Lobatae both well-drained and wet soils forests are equally possible environments. You may want to update to the current classification of Quercus (Denk et al. 2017). We are going to update the classification of Quercus, and discuss the different environments in the corresponding section considering Denk et al. (2017). Again, we agree that many taxa can be assigned to two or more vegetation units.

General comment 5: page 7, line 29 Something is wrong here. The modern distribution of Fagus grandifolia is just exactly in E North America. Not precisely on the coast but close to it. Check in GBIF. See also Bennet 1985, J. Biogeography. I would expect fairly high percentages of Fagus pollen in a modern pollen diagram. Was Fagus less common during some of the time intervals investigated by you?

The related sentence is probably unclear, we will rephrase it according to T. Denk's comments. We agree, F. grandifolia is the only Fagus in North America (growing east of the continent). In our record, relative abundances of Fagus show an increasing trend towards the late mid Miocene. During the Burdigalian relative abundances reached 10% (unpublished data) and were even higher in the late mid Miocene (Prader et al.,

2017). We have discussed this in section 4.2. Please also compare our answer to Thomas Denk's comment below concerning subgenus Engleriana in this context.

General comment 6: relates to section 4.3 I must admit that I had hard times reading this part. p. 8, line 16: Which Pinaceae? On the pollen diagram I see Pinus and Cathaya – Cathaya is more like a mesophytic element and fits well into vegetation unit 4. Also consider that "several conifer taxa may also have been part of hammocks within peat-forming vegetation (e.g. Cathaya, Sequoia, Taiwania) and of raised bogs (Sciadopitys; Schneider, 1992; Dolezych & Schneider, 2007)." [from Denk 2016]. All in all, I am not convinced by the claimed correlation between glaciations and fluctuations in the palynological record investigated for this study. Again, I think the entire pollen profile must be presented, and if the authors want to make a case for shifts in some Pinaceae taxa reflecting cooling or warming then this should be illustrated based on a detailed pollen diagram.

We can follow several points of Thomas Denk and agree that this aspect needs a more detailed discussion.

p.8. line 16. Instead of speaking of "meso- and microthermal Pinaceae" we will change this into: "peaks of Pinus together with single pollen grains of Picea". The decision to include Cathaya into the Vegetation Unit 2 is connected to the previous investigation, which focused on vegetation fluctuations of the late Mid Miocene of the same area (Prader et al., 2017). We also want to mention that according to Liu and Basinger, (2000), Cathaya has had a wider ecological range during the Palaeogene and early Neogene, this has also been discussed in Prader et al. 2017.

Again, we also aim at a better comparability of both records (the one presented herein and in Prader et al. 2017). Similarly to Pinus, we also think that Cathaya would be more over-represented in our record if it was part of vegetation unit 4. As suggested above, however, we will discuss the taxa assignment in more detail and mention the aspect that Cathaya could also be part of vegetation unit 4. Similarly, we agree with T.

Denk that Pinaceae and Cupressaceae do not only grow in higher elevated areas or in lowlands. As mentioned above we are going to include a complete pollen diagram showing all taxa and discuss climatic changes such as cooling and warming in respect to specific taxa, not only vegetation units.

General comment 7: Aspects of the paper that are insufficiently addressed in the current version Biogeographic aspects could be discussed when making use of the full pollen diagram. For example, I noted that Eotrigonobalanus, "Eleagnus", and Cedrus provide links to Europe. Eucommia, Cathaya, Sciadopitys provide geographic links with East Asia but were widespread in the N Hemisphere during the Cenozoic and Cedrelospermum and Eotrigonobalanus are extinct taxa. Are these taxa represented in all time slices covered by this study, is there a pattern of extinction? How does the pollen assemblage studied here compare to the Brandon Lignite?

No, Cedrelospermum and Eotrigonobalanus are not represented in all time slides, (pollen grains similar to e.g. Eotrigonobalanus were found 15 times). Both extinct genera were still present in an unpublished dataset of the Burdigalian time interval. These taxa did not become extinct during the Oligocene.

We are going to expand the discussion about Eotrigonobalanus and are going to show its sporadic occurrence as well in the pollen diagram. We will discuss biogeographical links to Europe and Asia in the discussion section.

We did not compare the dataset with the Brandon Lignite because the age model says that the terrestrial record is rather a middle early Miocene than an very early Miocene record, but we agree that it is a good point to compare this record with the Brandon Lignite investigation as well as with the late Middle record of the New Jersey area (Prader et al., 2017).

I also noted that the Fagus pollen you figure is very interesting: it has a long and narrow colpus reaching almost to the poles of the grain. This is typical of subgenus Engleriana and the most distinct species Fagus grandifolia within subgenus Fagus.

[Figure]

We are thankful for this remark. We are going to discuss the topic Fagus subgenus Engleriana as suggested.

Specific comments. Throughout the text:

It is early Eocene, middle Miocene, etc. not Early Eocene, Middle Miocene

We are going to change these word to lower case.

Page 1, line 15. "altitudinal spatial and long-term temporal vegetation migration" Very much information in one sentence. I don't understand what you mean with "altitudinal spatial" Are you assigning taxa to vertical vegetation zones? Same with "long-term temporal". Perhaps better to just say long-term.

We agree that this specific sentence is too complicated, we will rephrase and probably divide the information into two sentences. In accordance with the comment below, we will replace the phrase "altitudinal spatial" with "topographic palaeovegetation movements" to be consistent with the sentence in line 18.

Page 1, line 18. "To infer possible topographic palaeovegetation movements" Does this mean the same as "altitudinal spatial" above. This is very confusing, please re-phrase.

Yes, we meant the same, see above.

In case of the following comment, we are going to change all sentences just as T. Denk recommended. Page 1, line 23. "Biotic responds to environment change" change to: Biotic responses to environmental change. Page 7, line 6. "rule out that this extinct lineage" change to: rule out that Trigonobalanopsis Page 7, line 14. "had a greater ecological range" change to: had a wider ecological range Page 7, line 14. "had a greater ecological range" change to: had a wider ecological Range

Page 7, line 29. "persistent" do you mean "common"? Yes, we meant common, we are going to change it respectively.

Page 7, line 31. "Contrary to Fagus and its spatiotemporal distribution, the Atlantic east

coast is currently a hot spot. . ." Please re-phrase. We will rephrase this in accordance with the comments above.

Page 8, line 1. "where Carya became the prevalent genus" Please re-phrase. We will rephrase this sentence to make it clearer.

Page 8, line 13. Do you want to say that the maritime setting of your sample sites buffered possible regional climate change. And that this could explain the weak signal in the palynological record? Yes. We conclude that this sentence also need rephrasing to state more clearly what is meant.

Page 8, lines 27, 28. Please re-phrase. We will rephrase this sentence.

Page 9, line 1. "might" or: is?

We meant might in this context.

Page 9, lines 17, 18. "a contrasted spatiotemporal distribution ..." This does not make sense, please re-phrase. What exactly do you want to say? Same for "enhanced floral turnover". Please re-phrase.

We will rephrase these sentences.

Page 9, lines 25 ff. Please re-phrase. And re-think the possible movements.

We agree that in context with the points discussed above, the conclusions need rephrasing. While we still think that the way we assigned taxa to palaeovegetation units in the MS is appropriate (if discussed and explained in more detail), we will alter the conclusions considering certain taxa. Concerning movements, we will add the aspect that we may not only have altitudinal movements of Pinus and other taxa, but also changes in the lower altitudes of hinterlands.

Figure 4. Oxygen isotope curve. You may want to colour warming and cooling trends using red and blue. This would make it easier to read the figure.

Thanks, we are going to include colours to represent warming and cooling trends.

Supplement, Plate S4-ii. "Eleagnus" looks similar to Boehlensipollis hohli from Rupelian to Aquitanian strata of France, Belgium, and Poland (Sittler et al., 1975; Stuchlik et al., 2014). Affinities are possibly with Eleagnus but possibly also with other genera. Still, you have a nice example of a European-E North American disjunction here.

We are thankful for this remark, we will go again into the identification and taxonomy of this pollen grain and we will change it in the following. As mentioned above we will discuss biogeographical links to Europe and Asia in the discussion section.

Literature cited Bennet, K., D., 1985. The spread of Fagus grandifolia across eastern North America during the last 18 000 years. Journal of Biogeography 12, 147-164. Denk, T. 2016. Palaeoecological interpretation of the late Miocene landscapes and vegetation of northern Greece: a comment to Merceron et al., 2016 (Geobios, doi:10.1016/j.geobios.2016.01.004). Geobios 49, 135-146. Denk, T., Grimm, G. W., Manos, P. S., Deng, M., Hipp, A., 2017. An updated infrageneric classification of the oaks: review of previous taxonomic schemes and synthesis of evolutionary patterns. In: Gil-Peregrin, E., Peguero-Pina, J.J., Sancho-Knapik, D. (eds) Oaks Physiological Ecology. Exploring the Functional Diversity of Genus Quercus. Tree Physiology 7, pp. 13-38. Springer Nature, Cham, Switzerland. Hooghiemstra, H., Lézine, A.-M., Leroy, S.A.G., Dupont, L., Marret, F., 2006. Late Quaternary palynology in marine sediments: A synthesis of the understanding of pollen distribution patterns in the NW African setting. Quaternary International 148, 29-44. Kotthoff, U., Müller, U.C., Pross, J., Schmiedl, G., Lawson, I.T., van de Schootbrugge, B., Schulz, H., 2008. Late Glazial and Holocene vegetation dynamics in the Aegean region: an integrated view based on pollen data from the marine and terrestrial archives. Holocene 18, 1019-1032. Larsson, L.M., Dybkjær, K., Rasmussen, E.S., Piasecki, S., Utescher, T., Vajda, V., 2011. Miocene climate evolution of northern Europe: A palynological investigation from Denmark. Palaeogeography, Palaeoclimatology, Palaeoecology 309, 161-175. Liu, Y.S., Basinger, J.F., 2000. Fossil Cathaya (Pinaceae) Pollen from the Canadian

High Arctic. International Journal of Plant Sciences 161, 829-847. McCartan, L., Tiffney, B.H., Wolfe, J.A., Ager, T.A., Wing, S.L., Sirkin, L.A., Ward, L.W., Brooks, J., 1990. Late Tertiary floral assemblage from upland gravel deposits of the southern Maryland Coastal Plain. Geology 18, 311-314. Prader, S., Kotthoff, U., McCarthy, F.M.G., Schmiedl, G., Donders, T.H., Greenwood, D.R., 2017. Vegetation and climate development of the New Jersey hinterland during the late Middle Miocene (IODP Expedition 313 Site M0027). Palaeogeography, Palaeoclimatology, Palaeoecology 485, 854-868. Sittler, C., Schuler, M., Caratini, C., Chateauneuf, J.J., Gruas-Cavagnetto, C., Jardine, S., Ollivier, M.F., Roche, E., Tissot, C., 1975. Extension stratigraphique, répartition géographique et écologie de deux genres polliniques paléogènes observés en Europe occidentale: Aglaoreidia et Boehlensipollis. Bulletin de la Société Botanique de France 122, 231-245. Stuchlik, L., Ziembi′nska-Tworzydło, M., Kohlman-Adamska ,A., Grabowska, I., Słodkowska B., Worobiec E., Durska, E., 2014. Atlas of pollen and spores of the Polish Neogene.Volume 4. Angiosperms (2). Kraków: W. Szafer Institute of Botany, Polish Academy of Science.